# Parathyroid Involvement by Thyroid Carcinoma: Results from a Single Institution and a Review of the Literature

**DOI:** 10.3390/medicina61050836

**Published:** 2025-05-01

**Authors:** Ran Hong

**Affiliations:** Department of Pathology, Medical School, Chosun University, 365 Philmundaero, Donggu, Gwangju 61453, Republic of Korea; nanih@chosun.ac.kr; Tel.: +82-62-230-6356; Fax: +82-010-9114-1960

**Keywords:** parathyroid gland, metastases, thyroid gland

## Abstract

*Background and Objectives*: Thyroid cancer (TC) is a common malignancy that accounts for approximately 1% of all human cancers. Given their anatomical proximity to the thyroid, the parathyroid glands (PTGs) are theoretically at risk of tumor involvement. However, PTG metastases are rare and may be underdiagnosed because of routine PTG preservation during thyroidectomy. This study aimed to identify cases of PTG invasion by TC through a 10-year retrospective review at Chosun University Hospital, along with an analysis of the existing literature. *Materials and Methods*: A total of 1032 thyroidectomy cases were reviewed, and PTG involvement was detected in 10 cases (0.97%). Clinicopathological characteristics were evaluated, and literature data were analyzed. *Results:* The affected patients included nine females and one male, with a mean age of 46 years (range: 25–77 years). Histological examination confirmed papillary thyroid carcinoma (PTC) in all cases. Tumor invasion into the perithyroid soft tissues was observed in nine patients, and central cervical lymph node metastases were present in four. All patients exhibited PTG Pattern A (direct invasion). *Conclusions:* Based on our findings and literature data, PTG involvement by TC has an incidence rate of 0.05–3.9%, predominantly affects women in their sixth to seventh decade of life, and appears to have no impact on prognosis unless accompanied by extensive extrathyroidal invasion. Further studies are necessary to determine whether PTG invasion should be integrated into the TNM staging system and to assess its prognostic and therapeutic implications.

## 1. Introduction

The thyroid gland produces hormones that control various bodily functions. Papillary thyroid carcinoma (PTC) is typically a slow-growing, less aggressive tumor that can spread to nearby lymph nodes and, in some cases, to other organs [1]. Owing to their anatomical proximity to the thyroid, the parathyroid gland (PTG) is expected to be associated with a high risk of tumor involvement in patients with thyroid cancer (TC). However, PTG involvement may be under-recognized because of routine efforts to preserve the glands during thyroidectomy to prevent postoperative hypocalcemia [2]. Despite the anatomical closeness of the thyroid and parathyroid glands, parathyroid involvement in TC remains poorly characterized and is often overlooked during routine pathological examination. Given the potential clinical implications—including surgical decision-making, accurate staging, and prognostic assessment—a systematic investigation of PTG involvement by TC is warranted. Therefore, we aimed to identify and analyze cases of PTG invasion in patients with TC. The exact incidence of PTG metastasis in TC remains unclear, because PTG infiltration is not routinely examined during surgery, even in cases of invasive tumors, and published reports on the involvement of PTG in TC are limited. Few reports on PTG invasion or metastasis from TC are available, and they focus primarily on PTC [3]. Parathyroid invasion by TC can occur via two distinct mechanisms. The most common route involves tumor infiltration through the thyroid capsule, leading to direct invasion of the PTG. Less frequently, hematogenous dissemination results in metastatic foci in the PTG [4]. We aimed to select cases with PTG infiltration from our cohort of patients with TC. Here, we present a series of patients who underwent thyroidectomy and were diagnosed with PTG invasion caused by TC. We also analyzed the associated clinical and pathological characteristics and reviewed the existing literature on this subject. This study was approved by the Ethics committee of Chosun University Hospital (Institutional Review Board of Chosun University Hospital, Gwangju, Korea), which waived the requirement for written informed consent owing to the nature of the study.

## 2. Materials and Methods

Over the 10-year study period (2015–2024), 1032 thyroidectomies were performed at the Chosun University Hospital in Gwangju, Korea. Among these patients, 10 had PTG metastases from TCs, and all of them had PTC. Clinicopathological data were collected for each patient, including sex, age at diagnosis, tumor size, histological type, tumor characteristics, and extent of metastatic disease in TC. All 10 patients underwent total thyroidectomy with lymphadenectomy of the central cervical compartment (level VI). All surgical specimens were processed in paraffin blocks and stored in the pathology department. For histological examination, 5-micron-thick sections were cut and stained with Hematoxylin and Eosin (H-E). The slides were then reviewed. Pathological staging was performed in accordance with the American Joint Committee on Cancer/International Union Against Cancer (AJCC/UICC) classification system [5]. Based on previous studies [2,3,6,7], PTG involvement by TC was classified into three patterns: (1) direct invasion: tumor infiltration from the primary TC directly into the PTG (Pattern A); (2) encapsulated extension: TC extends into the PTG, with cancer nests separated by an intervening fibrous capsule (pseudocapsule) (Pattern B); and (3) true metastasis: the presence of metastatic foci within the PTG, without any continuity from the primary TC (Pattern C).

## 3. Results

Of the 1032 cases of primary TC, 10 cases of concurrent PTG involvement were detected, accounting for 0.97% of the subjects affected by primary TC. This increased to 3.05% when looking at cases where the PTG were excised (10 of 328 cases). Table 1 summarizes the clinical, pathological, and follow-up data of the 10 patients included in the present study. Of these, 9 (90.0%) were female and 1 (10.0%) was male (female-to-male ratio 9:1), with median age of 45.9 years (range: 25–77 years). Histological examination revealed classic PTC in all patients. The tumor extended beyond the thyroid capsule invading the perithyroid soft tissues in 9 out of 10 cases; no patient exhibited massive extrathyroid extension (ETE) involving the larynx, trachea, or esophagus. Lymph node metastases in the central cervical compartment were observed in 4 of the 10 patients. PTG involvement in TC occurs via direct invasion (Pattern A) (Figure 1A). Patterns B and C were not observed. A final AJCC/UICC stage I was assigned to 7 of the 10 patients (aged <55 years), while stage II was assigned to 3 patients. After surgery, all the patients remained alive and showed no clinical or laboratory evidence of PTC recurrence. Table 2 presents the results of the literature review.

## 4. Discussion

### 4.1. The Rarity of Parathyroid Gland Involvement in Thyroid Carcinoma

Involvement of the PTG by TC is considered rare. Although PTG is anatomically adjacent to the thyroid gland, reports on its direct invasion or metastasis remain limited. Previous studies have estimated the incidence of PTG involvement by TC to range from 0.05% to 3.9% [2,7,8], with most cases associated with PTC. This low incidence may, in part, be due to the routine surgical preservation of the PTG to prevent postoperative hypocalcemia, leading to under-recognition of PTG invasion during thyroidectomy. PTG metastasis from TC is thought to occur via two main pathways [2]. The most common pattern of PTG involvement is tumor infiltration through the thyroid capsule, which leads to direct invasion of the adjacent PTG tissue. Although less frequently observed, histological evidence of hematogenous spread with metastatic foci has been reported [4].

### 4.2. Diagnostic Challenges and the Potential Underestimation of Parathyroid Invasion

The under-recognition of PTG invasion by TC is likely due to both diagnostic challenges and surgical practices that prioritize PTG preservation [2]. The reported frequency of PTG involvement in thyroid carcinoma likely does not reflect its true incidence. Even in cases of extensive TC, surgeons prioritize PTG preservation to minimize the risk of postoperative hypocalcemia, which may contribute to under-reporting of PTG involvement. Consequently, PTG involvement may be overlooked during thyroidectomy, leading to an underestimation of its actual prevalence [2]. Several studies have reported this phenomenon, with a higher prevalence in women. However, it is important to acknowledge that the sample sizes in these studies have been very limited, which may affect the reliability of this observation. Chrisoulidou et al. [2] reported a 0.5% incidence of PTG involvement in patients (10 out of 1770 cases). However, when analyzing cases in which the PTG was excised, the incidence increased to 3.94% (10 out of 254 cases). Similarly, Al Qahtani et al. [8] reported a 1.94% prevalence of PTG metastasis among all patients, which increased to 8.38% in patients who underwent PTG excision. In addition, Kakudo et al. [7] found an overall prevalence of 3.9%, which increased to 7.9% when only parathyroidectomy samples were analyzed. In the present study, of the 1032 cases of primary TC, 10 cases of concurrent PTG involvement were detected, accounting for 0.97% of the subjects affected by primary TC. This increased to 3.05% cases where the PTG was excised (10 out of 328 cases). These studies indicate that the prevalence of PTG involvement in TC is likely to be higher than that currently recognized. In many cases, metastatic disease in the PTG may persist in situ due to the routine preservation of PTG during thyroidectomy.

### 4.3. Clinical and Prognostic Significance of Parathyroid Invasion

Metastasis of primary thyroid cancer to the PTG remains poorly understood and is likely underdiagnosed, largely because of the standard practice of preserving PTG during thyroidectomy to reduce the risk of postoperative hypocalcemia [2]. The clinical impact of parathyroid metastases remains uncertain, especially when only a single gland is affected. Since serum calcium levels are primarily regulated by the remaining functional parathyroid glands, metastasis to one parathyroid gland is unlikely to cause significant physiological disturbances unless multiple glands are involved [2]. An unresolved question in the current literature is whether parathyroid invasion by TC should be considered a significant risk factor for recurrence and survival or if it is merely an incidental finding with no meaningful clinical or prognostic impact. Several studies have attempted to determine whether the presence of PTG metastases influences the outcomes of TC. Ito et al. [4] suggested that parathyroid gland invasion has minimal clinical significance or prognostic impact and proposed classifying it as T3a disease. However, in the present study, patients with pT4 disease had significantly worse disease-free survival (DFS) rates than those with pT3 disease, regardless of the presence of PTG involvement. Papi et al. [6] found that the most common pattern of spread was direct invasion, observed in 6 out of 10 cases. This finding is consistent with that of our patient, as evidenced by the pathological specimen. Extension of the primary tumor through an intervening pseudocapsule was also observed but was found to be a less common pattern of spread. One patient with an aggressive tall-cell variant of PTC was included. Additionally, lymph node metastases were identified in 3 of the 10 patients. Despite these findings, all patients remained disease-free, with undetectable serum thyroglobulin concentrations after a mean follow-up period of 5.6 years, which was an unexpected outcome. All patients were classified into the intermediate-risk category of disease recurrence. However, none of the patients in this study (*n* = 10) had detectable Tg levels after a mean follow-up period of 5.6 years. Al-Qahtani et al. [8] reported that PTG involvement in patients with differentiated thyroid carcinoma (DTC) is rare and is associated with older age, advanced disease stage, and ETE. The incidence of PTG involvement is high in the presence of ETE. They found DFS rates in patients with PTG metastasis to be similar to those in patients with T4a stage differentiated TC, with a higher risk of recurrence and therefore likely to benefit from postoperative radioactive iodine (RAI). Kakudo et al. [7] also found that PTG involvement was associated with worse DFS rates and was correlated with a more aggressive form of TC. PTG due to PTC was associated with either an advanced disease or a 10-year DFS rate of 80.8%, mainly in patients older than 55 years. PTG invasion was more commonly observed in older patients, and this group had a higher proportion of male patients compared to those without PTG invasion. These patients have a higher incidence of ETE and are more frequently diagnosed at advanced stages of the disease. Lung metastases were observed in 2 out of the 14 patients, with a significantly higher incidence than in the control cases. Moreover, male patients with PTG invasion and those older than 55 years had reduced DFS rates compared to patients without PTG invasion. Ito et al. [4] demonstrated that patients with massive ETE had a significantly worse prognosis, regardless of PTG involvement. In fact, TC extension to the PTG itself had no impact on the 10-year DFS rate, which exceeded 95% in cases where only minimal tumor extension was present. In multivariate analysis, additional predictors of poor prognosis, aside from massive ETE, included age more than 55 years, N1b stage disease, and primary tumor size greater than 4 cm. Based on these findings, the authors concluded that “extension to the PTG corresponds to minimal ETE for PTC”. The current staging system for DTC does not classify the PTG as a metastatic site [9]. Thus, they validated the appropriateness of pTNM parameters used by the American Joint Committee on Cancer (AJCC/UICC) classification system for thyroid malignancies [5]. Specifically, they supported the classification of tumors as pT3 when they extend beyond the thyroid capsule to invade structures such as the subcutaneous soft tissues, larynx, trachea, esophagus, or recurrent laryngeal nerve. Furthermore, in alignment with the current American Thyroid Association (ATA) guidelines [9], patients with TC exhibiting minimal extension to the perithyroid soft tissues and concomitant PTG metastasis should be classified as low-risk. This classification is consistent with the findings in our patient cohort. Further studies are needed to clarify whether PTG involvement warrants the upstaging of TC (e.g., T3a/T4a for direct invasion or M1 for hematogenous spread) [8,10]. Given these prognostic considerations, it is important to address appropriate postoperative management strategies, including the potential role of RAI therapy in cases of parathyroid involvement.

### 4.4. Potential Role of Radioactive Iodine Therapy for Parathyroid Involvement

Although the parathyroid glands themselves do not typically concentrate radioactive iodine, parathyroid involvement in thyroid carcinoma usually reflects ETE of the tumor. According to the ATA guidelines, ETE is considered an indicator of higher risk and warrants postoperative RAI therapy to improve disease control and reduce the risk of recurrence [9,10]. Therefore, in cases where parathyroid invasion by thyroid carcinoma is identified, RAI therapy is generally recommended as part of the standard postoperative management, following the protocols for high-risk differentiated thyroid carcinoma (DTC) [4,10]. However, further studies are needed to clarify the specific impact of parathyroid involvement on the efficacy of RAI treatment.

When PTG invasion is confirmed, the surgical approach must be discussed. Preservation of PTG is a crucial strategy for preventing postoperative hypocalcemia. However, when the PTG is invaded by cancerous tissue, it is crucial to determine whether complete resection is necessary. Especially in cases where TC has extensively spread to the adjacent soft tissues and lymph nodes, a more aggressive treatment strategy, including PTG resection, may be necessary. To date, studies have lacked clear criteria for defining the extent of tissue resection, including PTG, during thyroid cancer surgery. Additionally, research on whether PTG invasion impacts the indications for postoperative RAI therapy is limited.

## 5. Conclusions

PTG invasion by TC is rarely reported, and its true incidence may be underestimated. Based on the results of this study and a review of the existing literature, the incidence of PTG invasion in TC ranges from 0.05% to 3.9%, with a higher frequency observed in women in their 60s–70s. These studies suggest that, unless accompanied by extensive ETE, PTG invasion in TC does not have a direct impact on prognosis. However, as most studies had a limited number of cases, future multicenter and long-term follow-up studies are needed to define the actual incidence of PTG metastasis. Future research should address the following key questions regarding PTG invasion by TC: (1) whether PTG invasion should be incorporated into the TNM staging system; (2) which strategy—resection or preservation—is more beneficial when PTG invasion is suspected during thyroid surgery; and (3) the impact of PTG invasion on the prognosis and treatment response of TC.

## Figures and Tables

**Figure 1 medicina-61-00836-f001:**
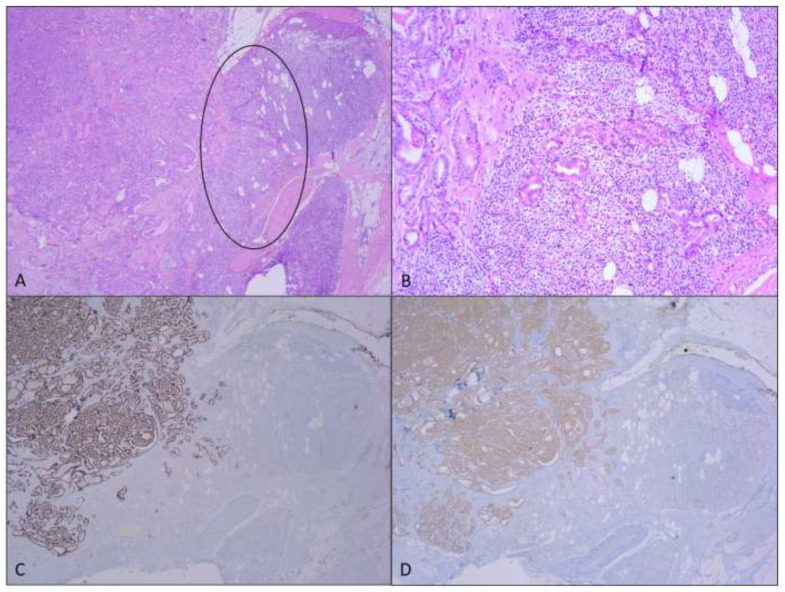
Representative histopathological findings from case #10, showing direct invasion of papillary thyroid carcinoma into the adjacent parathyroid gland (circle). (**A**) Hematoxylin and eosin staining, ×4, with highlighted area circled; (**B**) ×20. Tumor cells show immunoreactivity for CK19 and BRAF: (**C**) CK19, ×10; (**D**) BRAF, ×10.

**Table 1 medicina-61-00836-t001:** Clinical and histological characteristics of the 10 patients in the present study.

	Age	Sex	Histologic Type	Tumor Size	T Stage	AJCC Stage	ETE Other than PTG	LN Metastases	Pattern of PTG Metastasis
1	46	F	Classic PTC	0.7 × 0.7 cm	pT1a	I	x	x	Pattern A
2	48	F	Classic PTC	0.7 × 0.6 cm	pT1a	I	0	x	Pattern A
3	46	M	Classic PTC	0.7 × 0.5 cm	pT1a	I	0	x	Pattern A
4	29	F	Classic PTC	2.2 × 1.7 cm	pT2	I	0	+(8/44)	Pattern A
5	77	F	Classic PTC	1.5 × 1.2 cm	pT3b	II	0	+(10/22)	Pattern A
6	60	F	Classic PTC	3.4 × 6.4 cm	pT3a	II	0	x	Pattern A
7	25	F	Classic PTC	0.9 × 0.8 cm	pT1a	I	0	+(3/5)	Pattern A
8	40	F	Classic PTC	0.5 × 0.5 cm	pT1a	I	0	x	Pattern A
9	58	F	Classic PTC	0.8 × 0.7 cm	pT1a	II	0	x	Pattern A
10	30	F	Classic PTC	2.4 × 2.8	pT2	I	o	+(3/3)	II

F: female; M: male; ETE: extrathyroid extension; PTG: parathyroid gland; PTC: papillary thyroid carcinoma; LN: lymph node.

**Table 2 medicina-61-00836-t002:** Summary of the patients with parathyroid invasion by thyroid carcinoma reported in the literature.

Ref	Journal	Author	Year	N(%)	Mean Age	M/F Ratio	Stage	Type of TC	ETE	Distant Metastases	LN Metastases	PTG Invasion Pattern	Recurrence
[3]	Arch. Pathol. Lab. Med.	Tang et al.	2002	20 (2.2%)	52	6: 14	not reported	PTC-20	10	1	17	15: Pattern A 3: Pattern B 2: Pattern C	NR
[7]	J. Clin. Pathol.	Kakudo et al.	2004	14 (3.9%)	60.4	4: 10	I/II-2, III/IV-12	PTC-14	13	2 lung (2)	13	10: Pattern A 3: Pattern B 1: Pattern C	2
[4]	Endocr. J.	Ito et al.	2009	51 (1.6%)	NR	NR	pT3-30 pT4-21	Not reported	29	NR	NR	NR	1
[2]	World J. Surg. Oncol.	Chrisoplidou et al.	2012	10 (0.5%)	52.2	2: 8	I-2, II-1, III-5, IV-2	PTC-5, FVC-2 ATC-1	5	2 lung (1) Lung and brain (1)	5	9: Pattern A + B 1: Pattern C	NR
[6]	J. Thyroid. Res.	Papi et al.	2014	10 (0.05%)	55	3: 7	I-2, III-8	PTC-6, FVC-3 tall cell-1	6	0	3	6: Pattern A 2: Pattern A + B 1: Pattern B	0
[8]	J. Thyroid. Disorders. & Ther.	Al-Qahtani et al.	2014	16 (1.9%)	57.5	4: 12	pT2-2 pT3-6 pT4-8	PTC-15 FVC-1	13	5 lung (4) Bone (1)	14	14: PatternA 2: Pattern C	NR
Pre-sent				10 (9.7%)	46	1: 9	I-7, II-3	PTC-10	9	0	4	10: Pattern A	

NR: not reported [2,3,4,6,7,8].

## Data Availability

The datasets used and/or analyzed in the current study are available from the corresponding author upon reasonable request.

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
