# Peer review of "Parathyroid Involvement by Thyroid Carcinoma: Results from a Single Institution and a Review of the Literature"

_medicina, 2025, doi:10.3390/medicina61050836_

Round 1
Reviewer 1 Report
Comments and Suggestions for Authors
Dear Authors,
This study presents a very interesting paper on Parathyroid Involvement by Thyroid Carcinoma. The authors made a well structured presentation of used method and data presentation, as well as the detailed discussion on each question of importance. I consider your work an interesting contribution of the topic that should be accepted for the publication.
Author Response
Comment : This study presents a very interesting paper on Parathyroid Involvement by Thyroid Carcinoma. The authors made a well structured presentation of used method and data presentation, as well as the detailed discussion on each question of importance. I consider your work an interesting contribution of the topic that should be accepted for the publication.
Response : Thank you very much for your thoughtful and encouraging comments.
We are grateful for your positive evaluation of our study and your recognition of our work as a valuable contribution to the topic. We look forward to the next steps toward publication.

Reviewer 2 Report
Comments and Suggestions for Authors
The authors present a study on the importance of PTG involvement in DTC.
The paper seems interesting, well designed and presented. The sample is small, but justified by the rarity of the condition analysed.
My only suggestion is to clarify, perhaps with a note, the meaning of ">21" in Table 1 (ref "3") and to delete or adjust the unfocused part about the rarity of PTG metastases in the discussion.
Comments on the Quality of English LanguageLinguistic evaluation is not my area of expertise, so I suggest that you ask about it to another professionist.
Author Response
* Comment [1] My suggestion is to clarify perhaps with a note, the meaning of ">21" in Table 1 (ref "3")
Response[1]
I have corrected the part you pointed out “>21” (Tabe 2) to '29'. In the study by Ito et al. (ref.3), among 51 cases with parathyroid (PTG) involvement, 30 cases showed PTG involvement only, 21 cases showed involvement of both the PTG and other organs, and 8 cases involved the PTG and adjacent soft tissue. Based on this, I clarified the number of extrathyroid extension cases as “29”
* Comment [2] My suggestion is to delete or adjust the unfocused part about the rarity of PTG metastases in the discussion.
- Response[2]
In the initial part of the Discussion section titled 'The rare occurrence of malignant lesions and metastasis in the parathyroid gland', as you pointed out, the extended discussion of primary PTG cancers and metastatic lesions from other organs disrupted the logical flow of the manuscript. Therefore, I have deleted that portion and revised the text to focus exclusively on PTG involvement by thyroid carcinoma (TC).

Reviewer 3 Report
Comments and Suggestions for Authors
The authors have described cases of thyroid cancer with parathyroid involvement. The authors should include within their introduction and discussion the major sites and related routes of thyroid cancer metastasis.
The authors should discuss the potential of treatment of the extremely rare parathyroid involvement in thyroid cancer with therapeutic radioiodine.
The authors should provide an explanation within their introduction why they chose to study and investigate the possible involvement of parathyroids in the case of thyroid cancer.
Comments on the Quality of English LanguageEnglish should be improved.
Author Response
Comment [1] The authors should discuss the potential of treatment of the extremely rare parathyroid involvement in thyroid cancer with therapeutic radioiodine.
Response [1] Yes, I have addressed the point you raised by adding a subsection titled 'Potential role of radioactive iodine therapy for parathyroid involvement' at the end of the Discussion."
Comment[2] The authors should provide an explanation within their introduction why they chose to study and investigate the possible involvement of parathyroids in the case of thyroid cancer
Response [2] Yes, I have incorporated the point you raised into the middle part of the Introduction.

Round 2
Reviewer 3 Report
Comments and Suggestions for Authors
The authors have made the necessary amendments. The manuscript may now be published.